# AASeg: Attention Aware Network for Real Time Semantic Segmentation

## Abstract

In this paper, we present a new network named Attention Aware Network (AASeg) for real time semantic image segmentation. Our network incorporates spatial and channel information using Spatial Attention (SA) and Channel Attention (CA) modules respectively. It also uses dense local multi-scale context information using Multi Scale Context (MSC) module. The feature maps are concatenated individually to produce the final segmentation map. We demonstrate the effectiveness of our method using a comprehensive analysis, quantitative experimental results and ablation study using Cityscapes, ADE20K and Camvid datasets. Our network performs better than most previous architectures with a 74.4% Mean IOU on Cityscapes test dataset while running at 202.7 FPS.

## 1 Introduction

Semantic Segmentation is a fundamental problem in computer vision where the goal is to label each and every pixel in the image to its appropriate class. Since it is required to be deployed in real world settings like robots and autonomous vehicles, hence there is a need to balance the speed vs performance tradeoff. Fully Convolutional Network (FCN) (Long et al., 2015) composed of convolutional layers was one of the first works to get strong semantic representation.

However, this method was not able to capture boundary information accurately. Atrous convolutions (Yu and Koltun, 2015) at the last several stages of their network was used to give feature maps with strong semantic representation, thus solving the problem with FCN based architectures. However, this comes at at the cost of increased computational complexity. For real world navigation tasks like autonomous driving, there is a need to improve the Frame Per Second(FPS).

Chen et al. (2017) and (Yu and Koltun, 2015) used dilated convolutions to increase the recpetive field while maintaining the number of parameters. SegNet (Badrinarayanan et al., 2017) utilizes a small network structure and the skip-connected method to achieve improved FPS. (Mehta et al., 2018), (Paszke et al., 2016) and (Poudel et al., 2019) proposed unique approaches to tackle real time semantic segmentation problem.

## 2 Related Work

(Fu et al., 2019) introduces spatial-wise and channelwise attention modules to enhance the recpetive field (Paszke et al., 2016) trims a a lot of convolution filters to reduce computation. ICNet (Zhao et al., 2018a) proposed an image cascade network using multiresolution branches. (Iandola et al., 2016) allows the neural network to find the critical channels of the feature map and select the most suitable channels by itself.

ESPNet (Mehta et al., 2018) introduces an efficient spatial pyramid (ESP), which brings great improvement in both speed and performance. Bilateral Segmentation Network (BiSeNet) (Yu et al., 2018a) used two parts: Spatial Path (SP) is used to get with the loss of spatial information and Context Path (CP) for compressing the receptive field. (Yu et al., 2020) used multi-path framework to combine the low-level details and high-level semantics.

(Li et al., 2019b) utilizes a light-weight backbone to speed up its network and a multi scale feature aggregation to improve accuracy. SwiftNet (Orsic et al., 2019) used lateral connections to restore

the prediction resolution while maintaining the speed. (Lin et al., 2017) uses a multipath refinement network to refine the feature but ignores the global context feature.

The speed-Accuracy performance comparison of state of the art methods on the Cityscapes test set is shown in Figure 1:

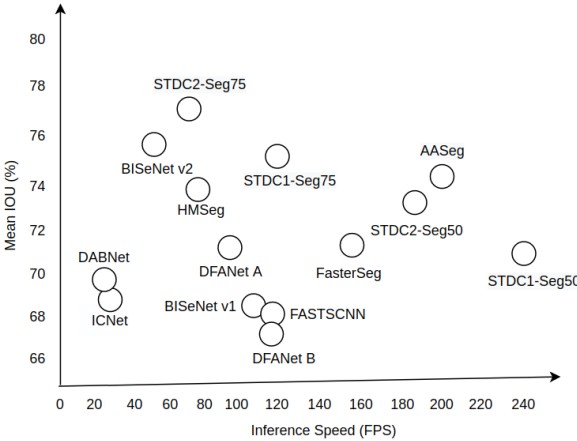

Figure 1: Speed-Accuracy performance comparison on the Cityscapes test set. Our approach achieves higher mean IOU while still being faster than most existing methods.
.

We summarize our main contributions as follows:

• We propose a novel Attention Aware Network (AASeg) for real time semantic segmentation.

• Our network is comprised of three parts: Spatial Attention(SA) module to capture the spatial dependencies of the feature maps, Channel Attention(CA) module to extract high level semantic information and Multi Scale Context(MSC) module to learn the information flow between feature maps of consecutive levels.

• Detailed experiments and analysis indicate the efficacy of our proposed network in not only improving the performance but also FPS. We achieve results on par with previous state of the art using Cityscapes, Camvid and on ADE20K datasets.

## 3 BACKGROUND

### 3.1 SPATIAL INFORMATION

The spatial information of the image is important to predict the detailed output for semantic segmentation. Modern existing approaches uses encoding of spatial information. DUC (Wang et al., 2018), PSPNet (Zhao et al., 2017), DeepLab v2 (Chen et al., 2017) use the dilated convolution to preserve the spatial size of the feature map.

### 3.2 CONTEXT INFORMATION

Semantic segmentation requires context information to generate a high-quality result. Most of the used methods enlarge the receptive field or fuse different contextual information. (Chen et al., 2017), (Wang et al., 2018) and (Yu and Koltun, 2015) use different dilation rates in convolution layers to capture different scale contextual information. In (Chen et al., 2017), an "ASPP" module is used to capture context information of different receptive field. PSPNet (Zhao et al., 2017) applies a "PSP" module which contains several different scales of average pooling layers.

### 3.3 ATTENTION MECHANISM

(Hu et al., 2018) applied channel attention for image recognition and achieve the state-of-the-art. (Yu et al., 2018b) proposed a network that learns the global context as attention and revise the features. Multi scale network along with a custom attention module was proposed by (Sagar and Soundrapandiyan, 2020) for semantic segmentation.

### 3.4 FEATURE FUSION

The spatial information captured by the Spatial Path consists of rich detailed information. The output feature of the Context Path is made up of contextual information. Feature fusion was used to achieve state of the art results on image classification, object detection and instance segmentation using DMSANet (Sagar, 2021b).

## 4 PROPOSED METHOD

### 4.1 DATASET

The following datasets have been used to benchmark our results:

**1. Cityscapes** It is used for urban street segmentation. The 5000 annotated images are used in our experiments which are divided into 2975, 500 and 1525 images for training, validation, and testing respectively.

**2. ADE20K** This dataset contains labels of 150 object categories. The dataset includes 20k,2k and 3k images for training, validation and testing respectively.

**3. CamVid** This dataset is used for semantic segmentation for autonomous driving scenarios. It is composed of 701 densely annotated images.

### 4.2 NETWORK ARCHITECTURE

The fundamental goal in semantic segmentation is to map an RGB image $X \in R^{H \times W \times 3}$ to a semantic map $Y \in R^{H \times W \times C}$ with the same spatial resolution $H \times W$, where $C$ is the number of classes of objects present in image. The input image $X$ is converted to a set of feature maps $F_l$ where l=1,...,3 from each network stage, where $F_l \in R^{H_l W_l C_l}$ is a $C_l$-dimensional feature map.

Our network dosen't use any backbone to extract features from the input image unlike many previous architectures. The input image is first passed through a block comprising of convolutional, batch normalization and ReLU activation function.

We express a convolution layer $W^n(x)$ as follows:

$$\mathrm{W}^n(x) = \mathbf{W}^{n \times n} \odot x + \mathbf{b} \tag{1}$$

where $\odot$ represents the convolution operator, $W^{n \times n}$ represents the $n \times n$ convolutional kernel, $x$ represents the input data and $b$ represents the bias vector.

### 4.3 MULTI SCALE CONTEXT MODULE

The feature map after passing through convolutional block is split to three different parts with $1 \times 1$ convolution, $3 \times 3$ convolution and $5 \times 5$ convolution respectively. The individual feature maps are then fused together. The output fused feature map is first convolved with a $1 \times 1$ convolution to reduce the number of channels from 2048 to 256. The feature map produced is of size $H \times W \times n_c$, where $H, W$ are the height and width of the feature map, and $n_c$ denotes the number of channels.

The input feature map is convolved with dilated convolution layers with increasing dilation rates of 3, 6 and 12. The dilated convolution layer input at every stage is formed by concatenating the input feature map with the outputs from previous convolutions. At the final step, the outputs from the three dilated convolutions are concatenated with the input feature map.

### 4.4 SPATIAL ATTENTION MODULE

The spatial attention module is used for capturing the spatial dependencies of the feature maps. The spatial attention (SA) module in our network is defined below:

$$f_{SA}(x) = f_{sigmoid}\left(W_2\left(f_{ReLU}\left(W_1(x)\right)\right)\right) \tag{2}$$

where $W_1$ and $W_2$ denotes the first and second $1 \times 1$ convolution layer respectively, $x$ denotes the input data, $f_{Sigmoid}$ denotes the sigmoid function, $f_{ReLU}$ denotes the ReLu activation function.

The spatial attention module used in this work is shown in Figure 2:

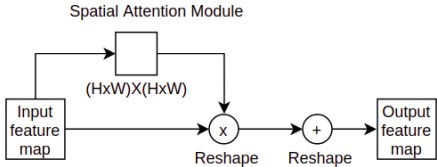

Figure 2: Details of our spatial attention module

.

### 4.5 CHANNEL ATTENTION MODULE

The channel attention module is used for extracting high level multi-scale semantic information. The channel attention (CA) module in our network is defined below:

$$f_{CA}(x) = f_{sigmoid}(W_2(f_{ReLU}(W_1 f^1_{AvgPool}(x)))) \tag{3}$$

where $W_1$ and $W_2$ denotes the first and second $1 \times 1$ convolution layer, $x$ denotes the input data. $f^1_{AvgPool}$ denotes the global average pooling function, $f_{Sigmoid}$ denotes the Sigmoid function, $f_{ReLU}$ denotes ReLU activation function.

The channel attention module used in this work is shown in Figure 3:

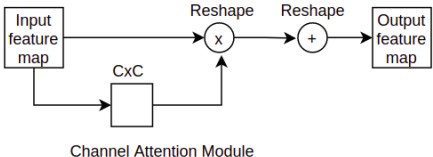

Figure 3: Details of our channel attention module

.

### 4.6 AGGREGATION

We denote the concatenation operation as follows:

$$x_{concat} = x_1 \oplus x_2 \oplus x_3 \tag{4}$$

where $\oplus$ represents the concatenation operator and $x_1$, $x_2$ and $x_3$ represents the features of the two branches. The AASeg module can be denoted as follow:

$$x_{AASeg} = ((f_{SA}(x_{concat}) \otimes x_{concat}) \oplus (f_{CA}(x_{concat}) \otimes x_{concat}) \oplus (f_{MSC}(x_{concat}) \otimes x_{concat})) \tag{5}$$

where $\oplus$ represents the concatenation operator, $f_{CA}$ represents the channel attention module mentioned in Equation 2, $f_{SA}$ represents the spatial attention module, $f_{MSC}$ represents the multi scale attention and $x_{concat}$ represents the combined feature.

we use $ConvX_i$ to denote the operations of $i^{th}$ block. Therefore, the output of $i^{th}$ block is calculated as follows:

$$x_i = ConvX_i\,(x_{i-1}, k_i) \qquad (6)$$

where $ConvX$ includes one of each convolutional layer, batch normalization layer and ReLU activation layer, $k_i$ is the kernel size of convolutional layer, $x_{i-1}$ and $x_i$ are the input and output of $i^{th}$ block. Fusion operation is used to combine high-level features with low-level features using Equation 7:

$$x_{output} = F\,(x_1, x_2, \ldots, x_n) \qquad (7)$$

The overall structure of our proposed AASeg network is shown in Figure 4.

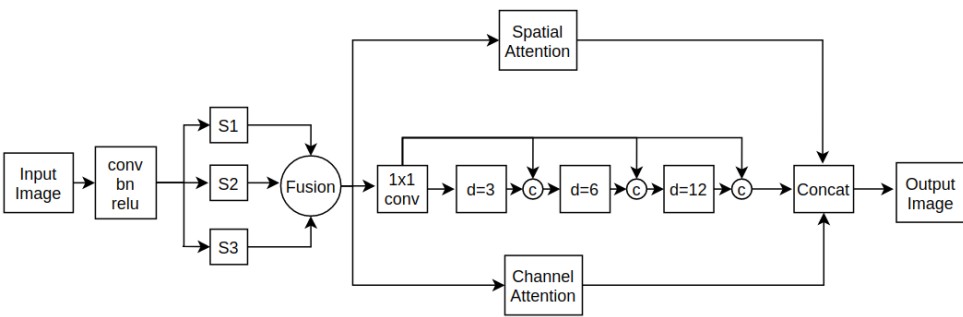

Figure 4: Architecture of proposed AASeg network architecture. "c" denotes concatenation.
.

## 4.7 LOSS FUNCTIONS

We use cross-entropy loss function to weigh the difference between the forward propagation result of the network and the ground truth of the samples. The cross-entropy loss is calculated as defined in Equation 8:

$$L_{ce} = \frac{1}{N} \sum_{n=1}^{N} [y_n \log \hat{y}_n + (1 - y_n) \log (1 - \hat{y}_n)] \qquad (8)$$

where $N$ denotes the total number of samples, $y_n$ denotes the probability that the forward propagation result is true, and $1\text{-}y_n$ denotes the probability that the forward propagation result is false.

We also use the auxiliary supervision loss $L_{aux}$ to improve the model performance thus making it easier to optimize. The auxiliary loss can be defined as:

$$\mathcal{L}_{aux} = -\frac{1}{BN} \sum_{i=1}^{B} \sum_{j=1}^{N} \sum_{k=1}^{K} \mathbb{I}\left(g_j^i = k\right) \log \left(\frac{\exp\left(p_{j,k}^i\right)}{\sum_{m=1}^{K} \exp\left(p_{j,m}^i\right)}\right) \qquad (9)$$

$$I\left(g_j^i = k\right) = \begin{cases} 1, & g_j^i = k \\ 0, & otherwise \end{cases} \qquad (10)$$

where $B$ is the mini batch size, $N$ is the number of pixels in every batch; $K$ is the number of categories; $p_{ijk}$ is the prediction of the $j^{th}$ pixel in the $i^{th}$ sample for the $k^{th}$ class, $I(g_{ij} = k)$ is a function which is defined in Equation 10.

The class attention loss $L_{cls}$ from channel attention module is also used. The class attention loss is defined as follows:

$$\mathcal{L}_{cls} = -\frac{1}{BN} \sum_{i=1}^{B} \sum_{j=1}^{N} \sum_{k=1}^{K} \mathbb{I}\left(g_j^i = k\right) \log \left(\frac{\exp\left(a_{j,k}^i\right)}{\sum_{m=1}^{K} \exp\left(a_{j,m}^i\right)}\right) \quad (11)$$

where $a_{ijk}$ is the value generated of the class attention map of the $j^{th}$ pixel in the $i^{th}$ sample for the $k^{th}$ class. We combine the three terms to balance final loss term as follows:

$$\mathcal{L} = \lambda_1 \mathcal{L}_{ce} + \lambda_2 \mathcal{L}_{cls} + \lambda_3 \mathcal{L}_{aux} \quad (12)$$

where $\lambda 1$, $\lambda 2$ and $\lambda 3$ are set as 1, 0.5 and 0.5 to balance the loss.

### 4.8 IMPLEMENTATION DETAILS

PyTorch deep learning framework is used to carry out our experiments. Our neural networks is trained with stochastic gradient descent (SGD) as optimizer, batch size of 16, momentum of 0.9 and weight decay of $5e^{-4}$. The network is trained for 20K iterations with an initial learning rate value of 0.01. The, "poly" learning rate policy is used to decay the initial learning rate while training the model to reduce over-fitting.

Data augmentation operations like random horizontal flip, random resizing with scale range of [1.0, 2.0], and random cropping with crop size of $1024 \times 1024$ was done.

### 4.9 EVALUATION METRICS

For quantitative evaluation of the performance of our network, mean of class-wise intersection-over-union (mIoU) is used for bench-marking the performance, number of float-point operations (FLOPs) and frames per second (FPS) are used for bench-marking the speed.

## 5 RESULTS

We present the segmentation accuracy and inference speed of our proposed method on Cityscapes validation and test set in Table 1. We use the training set and validation set to train our models before submitting to Cityscapes online server.

The qualitative results on Cityscapes validation set are presented in Figure 5:

Table 1: Comparisons with other state-of-the-art methods using Cityscapes dataset. no indicates the method do not have any backbone for training and testing. Best results are highlighted in bold.

| Model | Resolution | Backbone | mIoU val(%) | mIoU test(%) | FPS |
|---|---|---|---|---|---|
| DFANet B | 1024 × 1024 | Xception B | - | 67.1 | 120 |
| DFANet A | 1024 × 1024 | Xception A | - | 71.3 | 100 |
| BiSeNetV1 | 768 × 1536 | Xception39 | 69.0 | 68.4 | 105.8 |
| BiSeNetV1 | 768 × 1536 | ResNet18 | 74.8 | 74.7 | 65.5 |
| SFNet | 1024 × 2048 | DF1 | - | 74.5 | 121 |
| BiSeNetV2 | 512 × 1024 | no | 73.4 | 72.6 | 156 |
| BiSeNetV2-L | 512 × 1024 | no | 75.8 | 75.3 | 47.3 |
| FasterSeg | 1024 × 2048 | no | 73.1 | 71.5 | 163.9 |
| STDC1-Seg50 | 512 × 1024 | STDC1 | 72.2 | 71.9 | **250.4** |
| STDC2-Seg50 | 512 × 1024 | STDC2 | 74.2 | 73.4 | 188.6 |
| STDC1-Seg75 | 768 × 1536 | STDC1 | 74.5 | 75.3 | 126.7 |
| STDC2-Seg75 | 768 × 1536 | STDC2 | **77.0** | **76.8** | 97.0 |
| AASeg | 512 × 1024 | no | 74.8 | 74.4 | 202.7 |

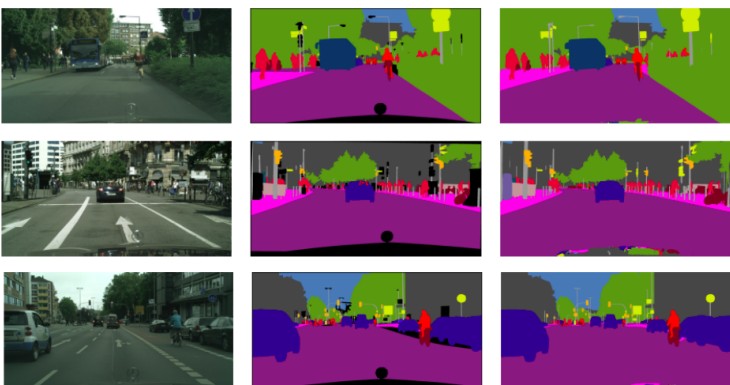

Figure 5: Visualized segmentation results on Cityscapes validation set. The three columns left-to-right refer to input image, ground truth, prediction from our network.
.

We evaluate our method on CamVid dataset. The comparison results with other methods is shown in Table 2:

Table 2: Comparisons with other state-of-the-art methods on CamVid. no indicates the method do not have a backbone for both training and testing. Best results are highlighted in bold.

| Model | Resolution | Backbone | mIoU(%) | FPS |
|---|---|---|---|---|
| DFANet A | $720 \times 960$ | no | 64.7 | 120 |
| DFANet B | $720 \times 960$ | no | 59.3 | 160 |
| BiSeNetV1 | $720 \times 960$ | Xception39 | 65.6 | 175 |
| BiSeNetV1 | $720 \times 960$ | ResNet18 | 68.7 | 116.3 |
| BiSeNetV2 | $720 \times 960$ | no | 72.4 | 124.5 |
| BiSeNetV2-L | $720 \times 960$ | no | 73.2 | 32.7 |
| SFNet | $720 \times 960$ | DF2 | 70.4 | 134.1 |
| SFNet | $720 \times 960$ | ResNet-18 | 73.8 | 35.5 |
| SFNet | $720 \times 960$ | DF2 | 70.4 | 134.1 |
| SFNet | $720 \times 960$ | ResNet-18 | 73.8 | 35.5 |
| STDC1-Seg | $720 \times 960$ | STDC1 | 73.0 | **197.6** |
| STDC2-Seg | $720 \times 960$ | STDC2 | **73.9** | 152.2 |
| AASeg | $720 \times 960$ | no | 73.5 | 188.7 |

The comparison of our network with previous state of the art methods for semantic segmentation using ADE20K validation set is shown in Table 3:

Table 3: Performance comparison with results reported on ADE20K validation set. Best results are highlighted in bold.

| Method | Mean IoU(%) | GFLOPs |
|---|---|---|
| PSPNet50 | 42.78 | 167.6 |
| PSPNet101 | 43.29 | 238.4 |
| SFNet | 42.81 | **75.7** |
| SFNet | 44.67 | 94.0 |
| DCANet | 45.49 | - |
| AASeg | **46.29** | 80.26 |

## 6 ABLATION STUDY

We explore the effect of 3 different upsamling operations: bilinear upsampling, deconvolution and nearest neighbor upsampling as shown in Table 4. The best results are obtained using bilinear upsampling but there isn't considerable difference.

Table 4: Ablation study on Upsampling operation in our network using Cityscapes validation set. Best results are highlighted in bold.

| Method | mIoU (%) |
|---|---|
| bilinear upsampling | **79.2** |
| deconvolution | 78.5 |
| nearest neighbor | 78.3 |

We also try the various kernel size in Table 5. Larger kernel size of $7 \times 7$ is also tried which results in a similar performance but increases computational complexity.

We use increasing sequence of dilation rates in our network, which is 0, 1, 2, 3. It seems large dilation rates increases the performance but comes at the cost of increased number of parameters, hence decrease in FPS. This effect is shown in Table 6:

Table 5: Ablation study on kernel size k in our network using Cityscapes validation set. Best results are highlighted in bold.

| Method | mIoU (%) | GFlops |
|--------|----------|--------|
| k = 1 | 79.2 | **118.2** |
| k = 3 | 79.4 | 120.8 |
| k = 5 | 79.5 | 127.5 |
| k = 7 | **79.5** | 136.1 |

Table 6: Ablation study results on Cityscapes validation set. FPS are estimated for an input image of resolution of $512 \times 1024$. Best results are highlighted in bold.

| Model | mIoU (%) | FPS | Parameters (M) |
|-------|----------|-----|----------------|
| AASeg-baseline | 79.2 | 118.2 | **0.76** |
| AASeg-w/o dilation | 77.4 | **121.3** | 0.80 |
| AASeg-(r=1) | 79.8 | 115.5 | 0.79 |
| AASeg-(r=2) | 80.1 | 102.7 | 0.84 |
| AASeg-(r=3) | **80.2** | 88.6 | 0.90 |

## 7 CONCLUSIONS

In this paper, we present a new network named Attention Aware Network (AASeg) for real time semantic segmentation. We use Spatial Attention(SA) and Channel Attention(CA) modules to enhance features of objects without adding extra computational cost. The fused spatial-channel attention along with Multi Scale Context(MSC) module enables network to extract discriminative and robust features of targets or background. We test our network using Cityscapes, ADE20K and Camvid datasets. Our network balances speed-performance tradeoff at par with previous state of the art network architectures. In the future, we would like to use multi scale attention module for instance segmentation problem.

ACKNOWLEDGMENTS

We would like to thank Nvidia for providing the GPUs.

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
