# OpenReview forum: "AASEG: ATTENTION AWARE NETWORK FOR REAL TIME SEMANTIC SEGMENTATION"
_ICLR.cc/2022/Conference — ICLR 2022 Submitted_

### Official Review · Reviewer_og3N · 2021-10-27

**Correctness:** 2
**Technical Novelty And Significance:** 1
**Empirical Novelty And Significance:** 2
**Recommendation:** 1
**Confidence:** 5

**Main Review:**

This paper has some issues as follows:
(1)	The novelty is quite limited. This paper proposed an attention based method for semantic segmentation, but the proposed spatial attention module and channel attention module are same with [1], and they are widely used in semantic segmentation.
(2)	Organization. The organization of this paper should be improved. The contribution of this paper should be in the Introduction section, rather than the related work. The introduction of datasets should be the in experiments section, rather than method section. It will be better to put the section of background in related work.
(3)	The motivation of this work is unclear. The authors should clarify the motivation in the introduction section.
(4)	The draws of the existing methods should be analyzed in the related work section.
(5)	The authors claim that atrous convolutions are able to capture boundary information, why?
(6)	The proposed modules are not evaluated in ablation study.
(7)	Wrong claims: “DUC (Wang et al., 2018), PSPNet (Zhao et al., 2017), DeepLab v2 (Chen et al., 2017) use the dilated convolution…”, “Our network dosen’t use any backbone”
(8)	Bad representation: “It also uses dense local multi-scale context information using Multi Scale Context (MSC) module.”

[1] Dual Attention Network for Scene Segmentation, CVPR 2019


**Summary Of The Paper:**

This paper proposed an attention aware network for real time semantic segmentation. Spatial attention and channel attention modules are used in the proposed method, as well as a multi scale context module. The proposed method is evaluated on some public datasets.

**Summary Of The Review:**

Mainly because of the weak novelty of the proposed method and poor organization of the paper, I will reject this paper.

---

### Official Review · Reviewer_dXYM · 2021-11-01

**Correctness:** 2
**Technical Novelty And Significance:** 1
**Empirical Novelty And Significance:** 1
**Recommendation:** 1
**Confidence:** 4

**Main Review:**

The paper is not well written. It feels like more of the school project report. It doesn’t explain what are the novelties and why the current approach outperforms other competing methods.

The related works are incomplete and mostly outdated. Between Intro and Related Works there is only a citation to a 2020 segmentation approach. Also, there are many references at the bottom that are not cited in the paper.

The background section quickly summarizes other segmentation works. However, it does not explain what distinguishes these approaches from the proposed method.

None of the components like architecture and losses is novel or used in a novel way. The combination thereof is also not novel.

The analysis lacks comparisons with other approaches like TDNet that is using channel attention.

**Summary Of The Paper:**

The paper proposes an approach to perform video semantic segmentation using the established non-local attention mechanism to fuse spatio-temporal features.

**Summary Of The Review:**

Reject due to lack of novelty, claims not explained and supported.

---

### Official Review · Reviewer_2HPE · 2021-11-02

**Correctness:** 3
**Technical Novelty And Significance:** 2
**Empirical Novelty And Significance:** 2
**Recommendation:** 3
**Confidence:** 4

**Main Review:**

Strength
1. The idea to omit heavy backbones is worth exploring and the proposed model manages to achieve a good performance with a lightweight architecture.
2. The models runs fast and addresses the need of real-time application.

Weakness
1. The submodules are not well explained. For example, there are addition ops in Fig.2&3, but they only take one input so I don't understand how the addition is done. The reshape in addition to the addition is also unclear.
2. The motivation of the proposed loss is unclear.
The CE loss (Eq. 8) is a binary cross entropy loss, where the groundtruth is "whether the forward propagation result is true". I don't quite understand what is "forward propagation result" for a semantic segmentation task.
Eq.(9), I believe, is in fact the standard pixel-wise multi-class cross entropy loss for semantic segmentation tasks and the authors call it "auxiliary" loss, which I find very confusing.
Eq.(10) is a loss applied to "the class attention map", which is not defined in previous sections. I assume it's channel attention from Sec. 4.5. Eq.(10) and Eq.(9) are essentially the same, but applied to the outputs from different modules.  Therefore, from my perspective, the "class attention map" can be treated as class predictions of each pixel, but it's called "attention", which seems confusing, too.
3. The ablation studies are poorly designed.
The first ablation is about upsampling op, which is not discussed in previous manuscript. I don't even know where those upsampling ops are in the whole architecture.
I expect the authors to ablate the three components, MSC, SA, CA, so that we can understand the contribution from each module, but only some detailed designs, such as kernel sizes and dilation rates, are ablated. In Table 6, the baseline architecture is unclear.
I also expect the authors to oblate the losses, e.g. what if there's no CE loss (Eq.8).
4. Comparing with previous real-time methods, the performance gain is marginal.
5. Limited novelty. Attention-based modules and skipping heavy backbones are not new for segmentation tasks.

**Summary Of The Paper:**

This paper proposes a real-time semantic segmentation. To save computation, there is no heavy backbone to extract features but just simple block containing conv/relu/bn. Then there three modules applied in parallel on top of the image features: multi-scale context module (conv with different dilation rate), spatial attention module and channel attention module.
Experiments were done on three datasets and the proposed method can achieve a reasonable performance with higher FPS than previous methods.

**Summary Of The Review:**

The paper made some contribution by proposing a lightweight architecture and achieving real-time inference.
However, the design is not well explained (figures and texts sometimes don't match). The experiments didn't explain the function of each module and focused on too-low-level ops. The performance gain and speed gain are marginal comparing with other real-time methods.

---

### Official Review · Reviewer_kwfp · 2021-11-03

**Correctness:** 2
**Technical Novelty And Significance:** 1
**Empirical Novelty And Significance:** 1
**Recommendation:** 1
**Confidence:** 5

**Main Review:**

1. Strengths:

(1) The network is simple and effective. The achieved trade-off between accuracy and inference speed is impressive.

2. Weakness:

(1)  The motivation is not clear. The introduction is so short, which didn't illustrate the clear motivation.

(2) The novelty is limited. The attention module is over-explored in this community. The proposed network has nothing new. The main structures of Fig.2 and Fig.3 has no difference with DANet. The multi-scale context module is like the DenseASPP. The simple combination of the off-the-shelf technologies far from reaches the level of ICLR.

(3) The gaps of mIoU and FPS between Tab. 1(test set) and Tab. 6 (validation set) are too large, which are beyond our common sense. Please  give detailed reasons.

(4) In the spatial/channel attention module, Eq. 2 and Eq. 3 indicate that the input features simply passed through two 1x1 convolutions. Why the output shape of Eq. 2 is (HxW)x(HxW)? Why the output shape of Eq. 3 is CxC?

(5) "x1, x2 and x3 represents the features of the two branches." Two branches produce three features?

(6) Please refine the typesetting. Page 7 has too much blank space.

(7) Why the performance on ADE20K dataset is so high? Did you pre-train on the ImageNet dataset?

(8) "Our network dosen’t use any backbone to extract features from the input image unlike many previous
architectures. " This claim is wrong. What's the meaning of the backbone? Could your network be called a backbone? Maybe you mean the pre-trained model.

(9) The whole organization is poor.
+ Please give the rigorous formulations for the spatial/context information, attention mechanism, feature fusion. Or move the background part to the related work.
+ Please summarize different methods in the related work, instead of just enumerating them.
+ Please move the section 4.1 (dataset part), 4.8 (implementation details), 4.9 (metrics) to the experiment part. They are not related to the method part directly.
+ The section 4.2 "network architecture" doesn't illustrate the whole architecture, just introduce a convolution layer.
+ Figure 4 can be illustrated in Section 4.2.

(10) The English writing is also poor.
+ Chaotic tense.
+ The second paragraph demonstrates the dilated convolution increases the computation complexity, which motivated improving the model speed. However, the third paragraph still introduces the dilated convolution method. The logic doesn't make sense.
+ The second paragraph introduce that "(Mehta et al., 2018), (Paszke et al., 2016) and (Poudel et al., 2019) proposed unique approaches". What are the unique approaches? Please summarize them.
+ "and ON ADE20K dataset" in the second contribution
+ The "use" is USED too much in the whole manuscript, which can be replaced by "perform", "conduct", "apply", "adapt", etc.
+ The symbol of "l" in Line 4, Paragraph 1, Section 4.2 is not correct.
Typos:
+ Duplicated "at" in Line 4 of the second paragraph
+ The format of the reference of "Chen et al. (2017) " in the third paragraph is incorrect.
+ The comma is required before "respectively".



**Summary Of The Paper:**

This manuscript proposes an Attention Aware Network for real-time semantic segmentation. This network is designed from scratch, which contains a spatial attention module, a channel attention module and a multi scale context module. It achieves an impressive tradeoff between accuracy and inference speed on three datasets. However, this manuscript seems like an unfinished tech report. The motivation is not clear. The proposed method is not novel. The details are missing. The writing and organization are poor. There are so many problems in this manuscript.

**Summary Of The Review:**

This manuscript needs careful refinement. This version far from reaches the level of ICLR. Maybe the authors can improve it according to the weakness part.

---

### Decision · Program_Chairs · 2022-01-20

**Decision:**

Reject

**Comment:**

All reviewers recommended reject, and there were no responses from authors.